# The Effects of Logistics Websites' Technical Factors on the Optimization of Digital Marketing Strategies and Corporate Brand Name

**Damianos P. Sakas** [1], **Dimitrios P. Reklitis** [1,*], **Panagiotis Trivellas** [2], **Costas Vassilakis** [3] **and Marina C. Terzi** [1]

1   Business Information and Communication Technologies in Value Chains Laboratory (BICTEVAC LABORATORY), Department of Agribusiness and Supply Chain Management, School of Applied Economics and Social Sciences, Agricultural University of Athens, 11855 Athens, Greece; d.sakas@aua.gr (D.P.S.); marinaterzi@aua.gr (M.C.T.)
2   Organizational Innovation and Management Systems (ORIMAS LABORATORY), Department of Agribusiness and Supply Chain Management, School of Applied Economics and Social Sciences, Agricultural University of Athens, 11855 Athens, Greece; ptrivel@aua.gr
3   Department of Informatics and Telecommunications, Akadimaikou G.K. Vlachou, 22100 Tripoli, Greece; costas@uop.gr
*   Correspondence: drekleitis@aua.gr; Tel.: +30-694-524-0492

**Abstract:** In a world overwhelmed with unstructured information, logistics companies increasingly depend on their websites to acquire new customers and maintain existing ones. Following this rationale, a series of technical elements may set the ground for differentiating one logistics website from another. Nevertheless, a suitable digital marketing strategy should be adopted in order to build competitive advantage. In this paper, the authors attempt to respond by implementing an innovative methodology building on web analytics and big data. The first phase of the research collects data for 180 days from 7 world-leading logistics companies. The second phase presents the statistical analysis of the gathered data, including regression, correlations, and descriptive statistics. Subsequently, Fuzzy Cognitive Mapping (FCM) was employed to illustrate the cause-and-effect links among the metrics in question. Finally, a predictive simulation model is developed to show the intercorrelation among the metrics studied as well as various optimization strategies. Research findings reveal a significant correlation between the logistics websites' technical factors and the growth of the corporate brand name.

**Keywords:** big data; web analytics; logistics; digital marketing; advertising; predictive model; brand name; user engagement; SEM; competitive advantage

## 1. Introduction

### 1.1. Digital Marketing in Logistics Companies

According to previous research, the total size of the logistics industry in 2018 was USD 5.58 trillion, and it has been predicted that in 2024 will be USD 6.88 trillion worldwide [1]. This prediction illustrates a 12.3% increase in the total market, and the main questions that the marketers and strategists must respond to are the following. (a) How can logistics firms gain more market share in this highly competitive environment? (b) What marketing strategy needs to be followed in order to maintain the existing customers and attract new ones faster? (c) Considering those predictions, digital marketing can highly contribute to the growth of logistics companies by analyzing the web analytics that plays a crucial role in building a competitive advantage [2,3]. In this paper, the authors attempt to answer these questions based on big data and web analytics and to examine websites' technical factors as well as users' behavior in the websites. Previous studies advocated that logistics companies need to develop their websites in an easy-to-use and user-friendly

way to attract new consumers and maintain the existing ones [4,5]. A well-designed website, in addition to the user experience benefits, also provides an opportunity for better implementation of the corporate digital marketing strategy [6].

Since one of the main forces widening the gap between markets and businesses is the internet [7], digital marketing may play a crucial role in filling this gap [8]. According to previous research, digital marketing contributes to logistics companies' competitive advantage since its adoption is more economical and more targeted than conventional marketing methods [8,9]. For example, if a television campaign costs USD 40.00 per month, the Search Engine Marketing (SEM) campaign costs approximately 100 times less to attract the same number of customers [10]. Another interesting example is the comparison between the placed advertisements in newspapers and digital advertisements. To reach the same number of customers, newspapers' advertisements are 1000% more expensive in comparison to digital ones [11]. Finally, one of the main advantages of the adoption of a well-designed digital marketing strategy is "Interactivity" [8,12], which is the company's ability to create strong communication paths between the entity and the customers [13]. An increase in interactivity leads to an increase in user engagement between the corporate website and the clients and also leads to increased sales and brand name recognition [13,14]. The following subsections discuss how interactivity can be optimized with the accurate analysis and implementation of the website's big data and web analytics [15,16].

This paper is divided into five sections: Section 1 depicts the introduction and literature review; Section 2 defines the materials and methods used in this study; Section 3 presents the results of descriptive statistics, correlation, and regression analyses, as well as the Fuzzy Cognitive Map (FCM) and the Agent-Based Model (ABM) approaches. Next, Section 4 discusses the findings, followed by Section 5 with the conclusions, which include research and practical implications.

### 1.2. Big Data in Logistics Websites

Big data analytics is growing more popular in a variety of fields, from logistics [17] and marketing [18] to neuroscience [19] and psychology [20]. The reason behind that is the growing necessity of structuring and using all that unstructured information [21]. This logic came to the surface 2500 years ago when Thucydides, in his work "*Peloponnesian war*", explained that the person that possesses the ability to structure all the unstructured information can create intelligence, which leads to power [22]. Consequently, big data analytics can play a crucial role in corporate competitive advantage [23,24]. Big data may well be described as large volumes of complex data that need advanced tools and processes to gather and critically analyze for intelligence production [25,26].

According to previous researchers, four crucial elements can be found underlying big data [8,27]. Big data "Volume" refers to the volume and size of unstructured data that businesses gather and attempt to manage [8,25], for instance, the number of visitors that entered a website and the analysis of their behavior. Second, big data "Variety" refers to the variety and diversity of dissimilar and incompatible data types, such as raw data that need to be processed to provide an added value to the corporate marketing strategy [8,26,27]. Third, big data "Velocity" refers to the speed at which businesses collect and handle those data [8]. For instance, every 24 h, Google has to process 3,500,000,000 searches and Twitter 500,000,000 tweets [28]. Finally, big data "Value" from a corporate point of view is the most important one [25,27,29]. Big data's value is generally derived from pattern detection or an algorithm analysis, which leads to increased efficiency, greater customer interactivity, and other tangible and measurable benefits [25,27,29].

According to the above logic of big data, the following questions have emerged. What will happen if a logistics website does not process a parcel quote fast enough and the customer must wait? What will happen if the customer must wait too long for the webpage to fully load? To respond to these research questions, we have to estimate "how much time exactly" is the waiting period and why. With the valuable assistance of big data analytics, web analytics marketers and developers can respond to those questions and possibly revert

the unwanted outcomes. Consequently, the adoption of big data analytics is crucial for logistics companies to acquire a competitive advantage [30]. There are different types of big data analytics, such as audio analytics, social media analytics, video analytics, and predictive analytics [25,31–34]; this research is focused on predictive analytics. Predictive analytics refers to the extraction of historic unstructured data and the creation of predictions based on them [25,35–37]. Predictive analytics are useful in environmental research [38] and retailing [39] but are commonly used in the logistics industry also [40–44].

### 1.3. Web Analytics Key Performance Indicators (KPIs) and Corporate Brand Name

Our study is based on predictive analytics as well as their connection with logistics websites' web analytics. More specifically, that kind of analytics provides researchers and marketers the ability to gather logistics websites' historical data and then create a simulation to investigate the correlations between the examined metrics [37,39]. Another crucial factor that logistics companies have to take into consideration is the incorporation of web analytics into their marketing strategy to acquire a competitive advantage [2,45]. Web analytics (WA) can play a role in the acquisition of competitive advantage through the extraction and analysis of various metrics that contribute to a website's visibility and brand name, such as "Organic Traffic" and "Global Rank" [46–48]. Predictive analytics were obtained in the type of web analytics for this study since those were acquired from seven world-leading logistics webpages. The chosen companies are among the top 10 third-party logistics (3PL) companies based on their size and profit [49]. The main prerequisites were the use of Facebook and Instagram for their social media promotion as well as the use of their website for the digital promotion of their services. Following several research recommendations, the authors chose seven logistics websites, as that is considered an adequate number for the type of analysis that this research performs and depicts a sufficient number to produce results for the logistics sector [9,14].

The procedure of studying the behavior of users on a website is known as web analytics [2,3]. Previous studies have shown that logistics websites' brand name, as well as the digital marketing strategy, can be dramatically improved by the correct implementation of web analytics [4,5,9]. Key performance indicators are web analytics that have been extracted from the examined websites and have been utilized quantitatively [50]. The WA KPIs are divided into two categories; on the one hand, there are the technical factors which include "Fully Loaded Time", Total Page Size", and website's "Requests", and on the other hand, the behavioral KPIs, such as "Organic Traffic", "Average Duration", and "Pages per Visits" [4,51,52]. The authors extracted the behavioral KPIs from the SEMrush and Alexa platforms, and the technical factors were extracted from the GTmetrix platform. The examined metrics are presented in Table 1.

**Table 1.** Presentation of the extracted Web analytics KPIs.

| Web Analytics KPIs | Description of the Web Analytics KPIs |
|---|---|
| Organic Traffic | Organic traffic refers to users that arrive at the corporate website through a non-paid way [9,53,54]. |
| Fully Loaded Time | According to GTmetrix, Fully Loaded Time refers to the time in seconds that it takes for a website to fully load [55]. |
| Total Page Size | The sum of the totality of the components required to load a website is referred to as Total Page Size. This contains the HTML and CSS, as well as the pictures [56]. |
| Requests | The number of requests necessary to render a website is reduced as the number of elements on the page is reduced, resulting in quicker page loading [57]. |
| Global Rank | This WA Key Performance Indicator is derived from the overall traffic on the platform, including organic, social, and paid traffic. The lower the worldwide rank, the more well-known the website, since a website in the 1st place has a better ranking in comparison to a website in the 15th place [14,58]. |
| Bounce Rate | When a customer enters a website and immediately exits without seeing anything more, this is known as a bounce rate [59]. |

**Table 1.** *Cont.*

| Web Analytics KPIs | Description of the Web Analytics KPIs |
| --- | --- |
| Average Time on Site | This KPI measures how long a user remains on a corporate website [60]. |
| Pages per Visit | When users enter a corporate webpage, they view a number of pages; the KPI "Pages per Visits" calculates this number [61]. |
| Paid Traffic | Paid Traffic is generated solely through paid methods. For instance, when a user selects a Google ad and is redirected to the corporate website. [62,63]. |
| Social Traffic | When a user is redirected from Facebook, Instagram, or social media in general to the corporate website, it produces the KPI Social Traffic. [60,64]. This research is limited to Instagram and Facebook. |
| Total Visitors | This KPI calculates the total number of users that enter a corporate website each day [60,65]. |

*1.4. Research Hypotheses*

In order to build a competitive advantage, logistics companies have to analyze their environment through all the available sources, including digital ones. Among the digital sources that affect their strategy are the technical factors of the websites, which include "Fully Loaded Time", "Total Page Size", and "Request". Those factors are crucial since we can identify in a tangible quantitative view the efficiency of the website [4]. Additionally, the examination of the behavioral factors such as "Average Time on Site" provides added value to marketing strategy since those factors offer indirect information for the user experience [9,14]. The intercorrelated examination of both factors will present available digital marketing optimization strategies. The research gap can be located in the previously limited research in the field of digital marketing in the logistics sector with the assistance of big data and web analytics. Additionally, since it is crucial for logistics companies to acquire a competitive advantage, it is useful to incorporate big data in this analysis. The unique perceptiveness of this research is important since raw undisputed big data have been extracted and analyzed to provide useful results for the actual behavior of the users on these websites, which could provide valuable insights for the increase of the digital brand name in logistics companies.

The findings of this study will be useful for decision-makers, developers, and marketers to identify the specific technical elements that differentiate one logistics website from another as well as the best digital marketing strategy that should be adopted. More specifically, decision-makers can optimize digital investments by the examination of those web analytics and incorporating them into a general strategy to increase corporate brand name [66–68]. Additionally, website developers can get useful insights on why and how a user-friendly website can optimize visibility [4,69,70]. Finally, marketers can identify ways to accomplish competitive differentiation leading to a competitive advantage [71,72]. As a consequence, the following hypotheses have been generated to gain a better understanding of the significance of big data and WA implementation and their impact on the corporate brand name.

**Hypothesis 1 (H1).** *"Total Page Size" of logistics companies affects "Organic Traffic" variable through their "Total Visitors" metric.*

The first hypothesis attempts to identify if the technical parameter "Total Page Size" and the behavioral parameter "Total Visitors" affect the metric "Organic Traffic". This hypothesis aims to identify if the above technical factors affect logistics webpage visibility.

**Hypothesis 2 (H2).** *"Total Visitors" of logistics companies affects "Social Traffic" variable through their "Fully Loaded Time" metric.*

The second hypothesis focuses on the investigation of the effects of the behavioral parameter "Total Visitors" and the technical "Fully Loaded Time" with the metric "Social Traffic". This hypothesis attempts to identify the effects of the websites' technical factors on the traffic generated from social media platforms such as Facebook and Instagram.

**Hypothesis 3 (H3).** *"Requests" of logistics companies affect "Paid Traffic" variable through their "Bounce Rate" metric.*

This hypothesis attempts to determine if the technical parameter "Requests" and the behavioral parameter "Bounce Rate" affect the metric "Paid Traffic". This research hypothesis is important since logistics managers have to understand the effectiveness of their advertisements in correlation to the users that exit the website immediately after they enter.

**Hypothesis 4 (H4).** *"Average Visits Duration" of logistics companies affects "Paid Traffic" variable through their "Total Visitors" metric.*

The fourth hypothesis focuses on the effects of the parameters "Average Visits Duration" and "Total Visitors" on the "Paid Traffic". This hypothesis is crucial since the effectiveness of the behavioral metrics on paid advertisements can be identified.

**Hypothesis 5 (H5).** *"Global Rank" of logistics companies is affected by "Fully Loaded Time", "Total Page Size", and "Requests".*

The final hypothesis attempts to discover the effects of the logistics websites' technical factors "Fully Loaded Time", "Total Page Size", and "Requests" on the corporate brand name. The findings will provide interesting insights to developers and marketers about the usability and efficiency of corporate websites.

The purpose of the study is to examine the impact of the websites' technical factors on the corporate brand name. For this reason, five research questions have been formulated to provide a holistic assessment of the main research purpose. More specifically, it is crucial for the companies to know the impact of the website size and loading time on the website's visibility (H1, H2). Additionally, it is important to know the efficiency of their paid advertisements and how to optimize them (H3, H4), and finally, how all the previous elements combined affect the corporate brand name (H5).

## 2. Materials and Methods

In this study, the authors adopted an alternative methodology to evaluate the effects of the logistics websites' technical and behavioral factors on the corporate website visibility and brand name. This methodology was adopted since the raw extracted data are not affected by any potential cognitive bias [73,74]. At the first stage of the study, web metrics were gathered for seven world-leading logistics companies. Those data were gathered from internet platforms daily for 180 days for the behavioral and technical data, respectively. Additionally, statistical analysis was performed, more specifically, descriptive statistics, correlations, and regression analysis.

In the second part of the study, an exploration model was developed to present the intercorrelations between the examined metrics, as well as three optimization scenarios, to provide decision-makers with a clear picture of the examined metrics. Finally, after the macro-scale analysis, a microanalysis was implemented supported by regression and correlation analysis to create a simulation model that illustrates users' activity on a corporate website [75].

### 2.1. Selection, Retrieval, and Statistical Analysis

Web analytics were gathered from seven logistics websites. The logistics firms were selected based on their profitability [49,76]. SEMrush was used for the extraction of the

behavioral data and, more specifically, for the metrics: "Organic Traffic", "Global Rank", "Bounce Rate", "Average Time on Site", "Pages per Visit", "Paid Traffic", "Social Traffic", and "Total Visitors". GTmetrix was used for gathering technical metrics, such as: "Fully Loaded Time", "Total Page Size", and "Requests". After the extraction of data, the authors conducted a statistical analysis. This statistical analysis provides a comprehensive picture of all the factors and how those interconnected factors affect the marketing strategy.

### 2.2. Exploratory Model Creation

The statistical analysis revealed various correlations among the examined factors. The adoption of the Fuzzy Cognitive Map (FCM) is based on this statistical analysis. The FCM's main aim is to provide a graphic representation of the positive and negative, cause and effect connections between the web metrics that are being examined [75,77]. Additionally, the authors use this method to create three optimization scenarios since this macro-scale model was created to show the strength of correlations and can be utilized in the construction of a successful marketing strategy [75,77,78].

### 2.3. Agent-Based Model

After the creation of the macro-scale model (FCM), a micro-scale model was developed. Agent-Based Models (ABM) provide the ability to simulate and estimate all the behavioral and technical factors that affect the corporate brand name and visibility [9,75,79]. The contribution of the ABMs to digital marketing and decision-making is profound and has been analyzed in various previous research [80,81]. More specifically, ABMs provide the ability for marketers to create simulations for real-life problems and to analyze user behavior with no cost for the extraction of valuable decision-making strategies [79,81]. The following section presents the results of the study.

## 3. Results

### 3.1. Statistical Analysis

The results of the data gathering platforms, presented in Table 1, are discussed in this section. The results are based on data collected from seven world-leading logistics companies [76]. The collected data were merged per category to present the total results for the logistics sector. For instance, the metric "Webpages' Total Page Size" represents the analysis of all logistics companies. Table 2 illustrates the descriptive statistics from the data collection of 180 days.

**Table 2.** Descriptive Statistics for 7 logistics webpages during 180 consequent days.

|  | Mean | Min | Max | Std. Deviation |
|---|---|---|---|---|
| Webpages' Organic Traffic | 31,199,621.47 | 3,123,349.00 | 67,839,204.00 | 19,997,235.96 |
| Webpages' Paid Traffic | 445,757.35 | 11,435.00 | 1,564,843.00 | 502,902.50 |
| Webpages' Average Time on Site | 517.69 | 412.00 | 766.00 | 103.45 |
| Webpages' Bounce Rate | 0.468 | 0.349 | 0.592 | 0.079 |
| Webpages' Pages/Visit | 2.81 | 2.20 | 3.54 | 0.51 |
| Webpages' Total Visitors | 141,205,016.26 | 5,418,390.00 | 375,118,623.00 | 128,953,195.18 |
| Webpages' Global Rank | 11,944.90 | 8983.00 | 14,445.00 | 1900.42 |
| Webpages' Total Page Size | 1.947 | 0.874 | 9.198 | 1.0784 |
| Webpages' Requests | 90.80 | 27 | 152 | 24.983 |
| Webpages' Fully Loaded Time | 4.373 | 1.48 | 49.55 | 3.21452 |
| Webpages' Social Traffic | 1,410,459.42 | 17,309.00 | 3,837,538.00 | 1,431,734.58 |

Tables 3 and 4 illustrate Pearson's coefficients and regression analysis testing the first hypothesis (H1).

**Table 3.** Coefficients between the examined metrics for H1.

| Correlations | Organic Traffic | Total Page Size | Total Visitors |
|---|---|---|---|
| Organic Traffic | 1 | | |
| Total Page Size | 0.033 | 1 | |
| Total Visitors | 0.962 ** | 0.018 | 1 |

** Correlation is significant at the 0.01 level (1-tailed).

**Table 4.** First hypothesis' Regression.

| Variables | Standardized Coefficient | $R^2$ | F | p-Value |
|---|---|---|---|---|
| *Constant (Organic Traffic)* | - | 0.927 | 5357.912 | <0.001 |
| Total Page Size | 0.015 | | | 0.087 |
| Total Visitors | 0.962 ** | | | 0.000 |

** Correlation is significant at the 0.01 level (1-tailed).

As illustrated in Table 3, a significant positive correlation with $\rho = 0.962$ ** was observed between the traffic and Total Visitors, implying that as traffic increases, more people will visit the logistics webpage. Furthermore, non-significant correlations have been found between the total page size, the total visitors, and organic traffic with $\rho = 0.018$ and $\rho = 0.033$, respectively. This result illustrates that there is no correlation between the total webpage size and visibility. The regression analysis is presented in Table 4. The regression analysis model is significant, with *p*-values < 5%. The results are significant, and more specifically, with every 1% increase in organic traffic, the total visitors and total page size increase by 1.5% and 96.2% accordingly.

Tables 5 and 6 illustrate the Pearson's coefficients and the regression of the second hypothesis (H2).

**Table 5.** Coefficients between the examined metrics for H2.

| Correlations | Social Traffic | Fully Loaded Time | Total Visitors |
|---|---|---|---|
| Social Traffic | 1 | | |
| Fully Loaded Time | −0.029 | 1 | |
| Total Visitors | 0.931 ** | −0.012 | 1 |

** Correlation is significant at the 0.01 level (1-tailed).

**Table 6.** Second hypothesis' Regression.

| Variables | Standardized Coefficient | $R^2$ | F | p-Value |
|---|---|---|---|---|
| *Constant (Social Traffic)* | - | 0.868 | 2749.090 | 0.683 |
| Fully Loaded Time | −0.018 | | | 0.150 |
| Total Visitors | 0.931 ** | | | 0.000 |

** Correlation is significant at the 0.01 level (1-tailed).

As presented in Table 5, a significant positive correlation with $\rho = 0.931$ ** has been detected between the social traffic and total visitors, suggesting that as the social traffic increases, more people will visit the corporate website. Additionally, non-significant negative correlations have been found between the fully loaded time, the total visitors, and social traffic with $\rho = -0.029$ and $\rho = -0.012$, respectively. That indicates that when the webpage takes less time to load, more visitors access the website. This is the expected behavior since if the webpage takes too long to load, visitors exit the website [82]. As for the regression, with every 1% increase in Social Traffic, the total visitors increased by 93.1%, and fully loaded time decreased by 1.8%.

Tables 7 and 8 illustrate the Pearson's coefficients and the regression of the third hypothesis (H3).

**Table 7.** Coefficients between the examined metrics for H3.

| Correlations | Paid Traffic | Bounce Rate | Requests |
|---|---|---|---|
| Paid Traffic | 1 | | |
| Bounce Rate | 0.675 ** | 1 | |
| Requests | −0.013 | 0.024 | 1 |

** Correlation is significant at the 0.01 level (1-tailed).

**Table 8.** Third hypothesis' Regression.

| Variables | Standardized Coefficient | $R^2$ | F | p-Value |
|---|---|---|---|---|
| *Constant (Paid Traffic)* | - | 0.457 | 351.929 | <0.001 |
| Bounce Rate | 0.676 ** | | | <0.001 |
| Requests | −0.029 | | | 0.249 |

** Correlation is significant at the 0.01 level (1-tailed).

Table 7 illustrates a significant positive correlation with $\rho = 0.675$ ** between the paid traffic and the bounce rate. This is interesting since it indicates that most of the users that entered the websites from a paid advertisement exited from it without any activity. Additionally, non-significant correlations have been observed between the requests, the bounce rate, and social traffic with $\rho = 0.024$ and $\rho = -0.013$, respectively. The latter illustrates that paid advertisements, to be effective, need to be placed on a targeted crowd. The regression is presented in Table 8. Regression is significant, with $p$-values < 5%. The results of the H3 are significant, and with every 1% increase in paid traffic, the bounce rate increased by 67.6%, and requests decreased by 2.9%.

Tables 9 and 10 present the Pearson's coefficients and the regression of the fourth hypothesis (H4).

**Table 9.** Coefficients between the examined metrics for H4.

| Correlations | Paid Traffic | Total Visitors | Average Time on Site |
|---|---|---|---|
| Paid Traffic | 1 | | |
| Total Visitors | 0.766 ** | 1 | |
| Average Time on Site | 0.275 ** | −0.149 ** | 1 |

** Correlation is significant at the 0.01 level (1-tailed).

**Table 10.** Fourth hypothesis' Regression.

| Variables | Standardized Coefficient | $R^2$ | F | p-Value |
|---|---|---|---|---|
| *Constant (Paid Traffic)* | - | 0.742 | 1204.316 | <0.001 |
| Total Visitors | 0.826 ** | | | <0.001 |
| Average Time on Site | 0.398 ** | | | <0.001 |

** Correlation is significant at the 0.01 level (1-tailed).

As illustrated in Table 9, significant positive correlations with $\rho = 0.766$ ** and $\rho = 0.275$ ** were observed between the paid traffic, total visitors, and average time on site, respectively. This is as expected since paid advertisements increase the number of visitors and the time spent on site. On the other hand, a negative significant correlation was observed between the total number of visitors and the average time on site with =−0.149 **. This result indicates that the vast majority of the visitors who entered from a paid advertisement on the website might have entered accidentally and exited after some seconds. The regression

is presented in Table 10. Regression is significant, with *p*-values < 5%. More specifically, the results are significant since, with every 1% increase in paid traffic, the total visitors and average time on site increase by 82.6% and 39.8%, respectively.

Tables 11 and 12 illustrate the Pearson's coefficients and the regression of the fifth hypothesis (H5).

**Table 11.** Coefficients between the examined metrics for H5.

| Correlations | Global Rank | Fully Loaded Time | Total Page Size | Requests |
|---|---|---|---|---|
| Global Rank | 1 | | | |
| Fully Loaded Time | 0.088 * | 1 | | |
| Total Page Size | −0.268 ** | 0.059 | 1 | |
| Requests | 0.101 ** | −0.071 * | 0.158 * | 1 |

* Correlation is significant at the 0.05 level (2-tailed). ** Correlation is significant at the 0.01 level (1-tailed).

**Table 12.** Fifth hypothesis' Regression.

| Variables | Standardized Coefficient | $R^2$ | F | p-Value |
|---|---|---|---|---|
| *Constant (Global Rank)* | - | 0.107 | 33.291 | <0.001 |
| Fully Loaded Time | 0.117 ** | | | <0.001 |
| Total Page Size | −0.300 ** | | | <0.001 |
| Requests | 0.157 ** | | | <0.001 |

** Correlation is significant at the 0.01 level (1-tailed).

As illustrated in Table 1, multiple significant correlations were observed among global rank, fully loaded time, and requests with $\rho = 0.088$ *, $\rho = -0.268$ **, and $\rho = 0.101$ **, respectively, implying that as traffic increases, more users will visit the logistics webpage. Drawing from this finding, it can be seen that the global rank and, as a consequence, the brand name of a logistics website is affected by the technical factors of the website. Additionally, the regression is presented in Table 12. More specifically, the results are significant, and with every 1% increase in global rank, an increase can be observed in fully loaded time and requests as well as a decrease in total page size by 11.7%, 15.7%, and 30%, respectively. Given that $R^2$ is rather restricted (0.107), although this hypothesis is accepted, generalizations should be made with caution.

### 3.2. Fuzzy Cognitive Map

The above statistical analysis was used for the creation of the Fuzzy Cognitive Map. An FCM can illustrate the fundamental parameters of a system as well as present the level of correlations (−1,1) between the examined metrics [75,78]. This feature is important for decision-makers and useful for big data analysts. Since big data are unstructured information, the illustration on a map of the cause and effects relationships of a system is very useful for marketers [83]. This method was used in this paper since it was frequently used in earlier SEO and SEM studies [14,79,84,85]. Figure 1 presents the FCM with the degree of correlation. The thicker the line, the higher the correlation.

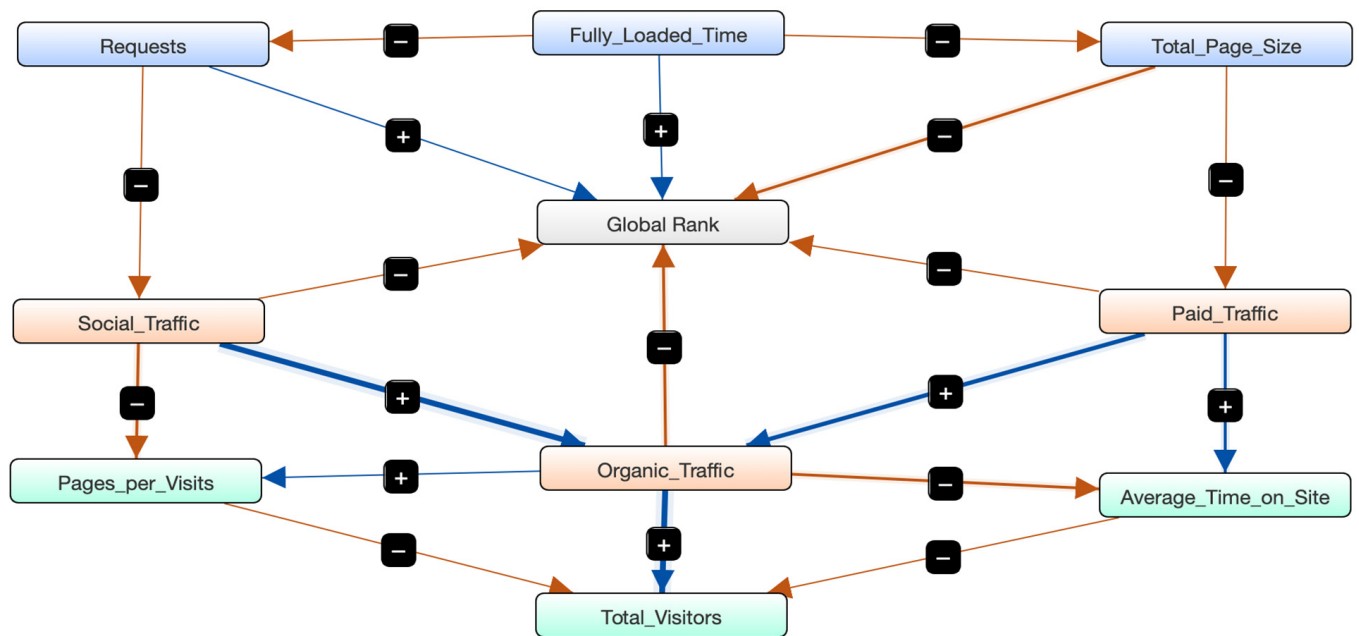

**Figure 1.** Fuzzy Cognitive Map illustration of the examined parameters.

### 3.2.1. Adoption of Fuzzy Cognitive Map Scenarios to Analyze the Data

Supporting the progress of the FCM model, three optimization scenarios were run to assess the predicted variations in the KPIs at different phases of user activity on the corporate website. In these scenarios, the Sigmoid function has been used because it provides the ability to use the interval $(-1,1)$ and illustrates the degree of impact between the examined metrics [86]. Figure 2 presents the technical factors optimization scenario. According to this scenario, if the company manages to reduce the loading time of the webpage by 5%, it will observe a 1% decrease in global rank, which is beneficial because the second website in rankings is better than the 15th, and the average time on site spent from a user by 1%. This is expected behavior because, according to previous research, a lot of users exit the website if a page takes more time than expected to load [82].

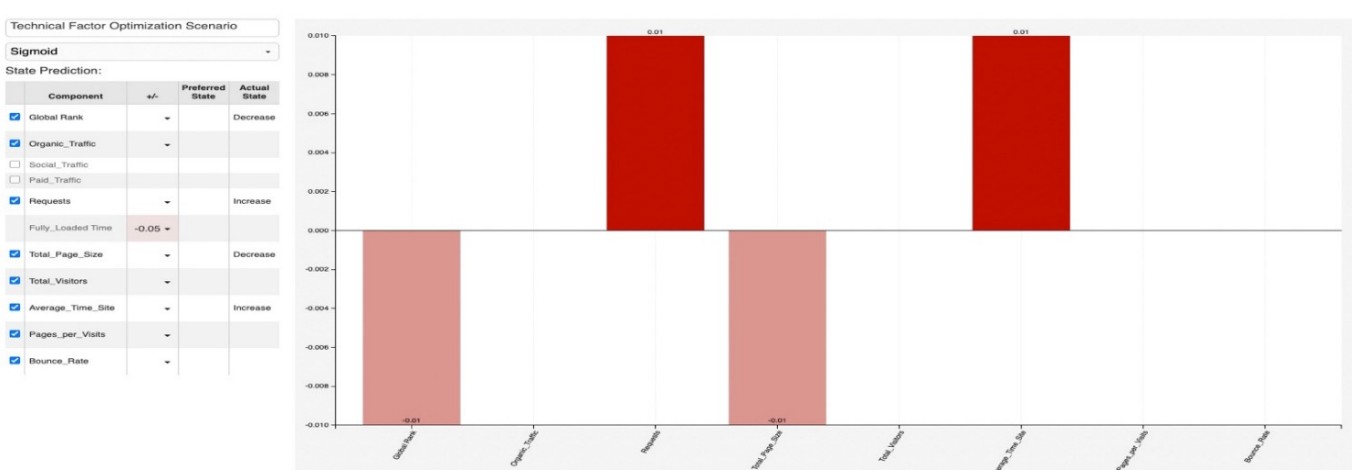

**Figure 2.** Technical Factors Optimization Scenario.

Figure 3 presents the visibility optimization scenario, and in this case, some interesting outcomes have been observed. If the company increases the visitors and the organic traffic by 10%, the total pages per visit will increase by 2%, the average time on site by 5%, and the bounce rate will decrease by 8%. This is expected because the sudden rise brings

the corporate website more users that spend more time and search more pages on the corporate website.

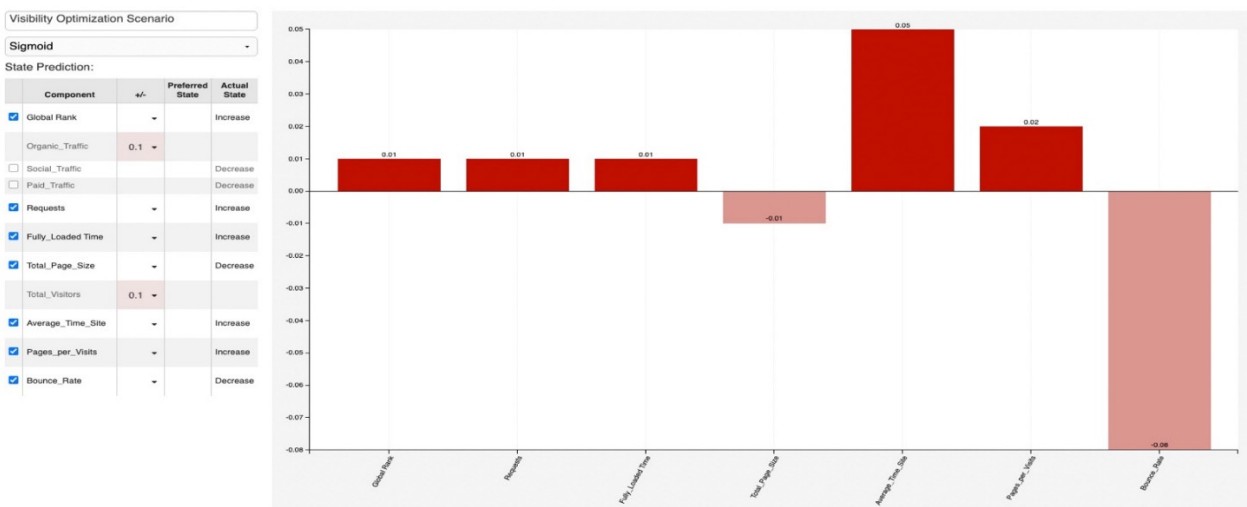

**Figure 3.** Visibility Factor Optimization Scenario.

Figure 4 presents the brand name optimization scenario. If the company's target is to decrease its global rank by 40%, it has to increase the organic traffic by 1%, the paid traffic by 1%, and the social media traffic by 3%. This interesting observation relies on the fact that social media advertisements have more impact on the logistics brand name than paid advertisements through search engines. Additionally, in order to acquire a better brand name, the logistics websites need to generate 1% more visitors well as decrease the fully loaded time by 2%.

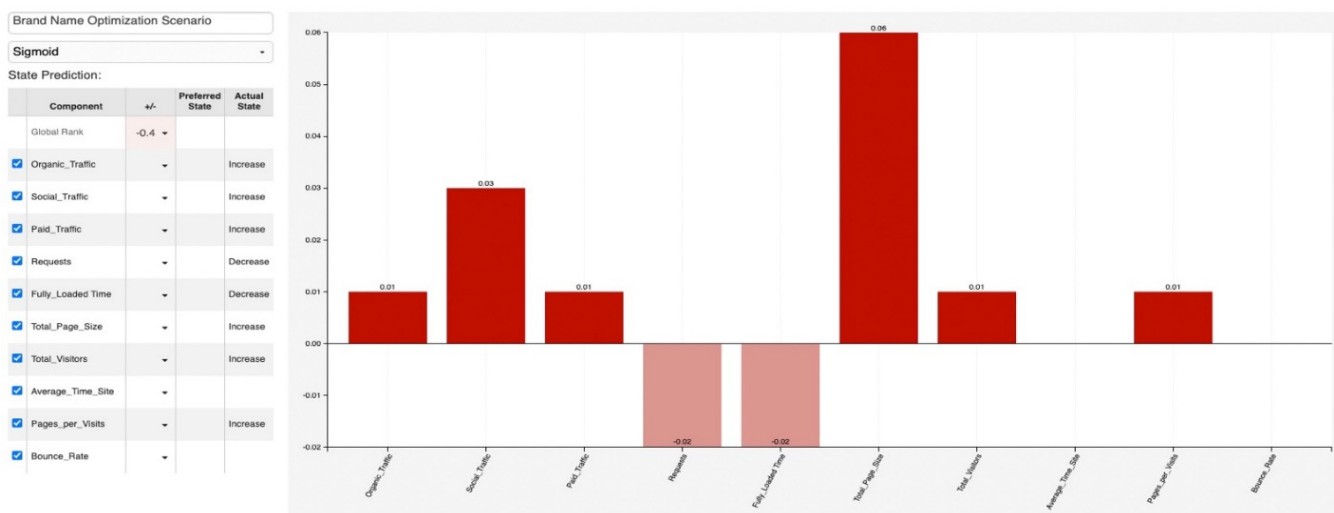

**Figure 4.** Brand Name Optimization Scenario.

### 3.3. Adoption of ABM

An agent-based model was adopted to provide a microanalysis of the issue [75]. Its adoption is beneficial for marketers and decision-makers since the agent-based models provide simulations for real-world situations and evaluate user behavior to extract useful decision-making and marketing strategies [9,79,81]. Additionally, the usage of Agent-Based Models allows organizations to fully comprehend the insights provided by big data in terms of user interaction with their websites, as well as chances for growth [79,81]. The

complete model was created using the Anylogic 8.7.9 software and the programming language "Java."

The model depicts the usual behavior of a customer that enters a logistics website. More specifically, the gray top box illustrates the initial position of an agent. The model simulates all the processes from the way that a customer enters the website (traffics) up to the production of the global rank. Changes to the needed circumstances are shown as variables (V) at the bottom of the image in Figure 5 that cause movement among the blocks, which is displayed on the ABM with black arrows. For this model, the Poisson distribution was used because it provides the ability to incorporate into the model the statistical analysis presented in Section 3.1 [87,88].

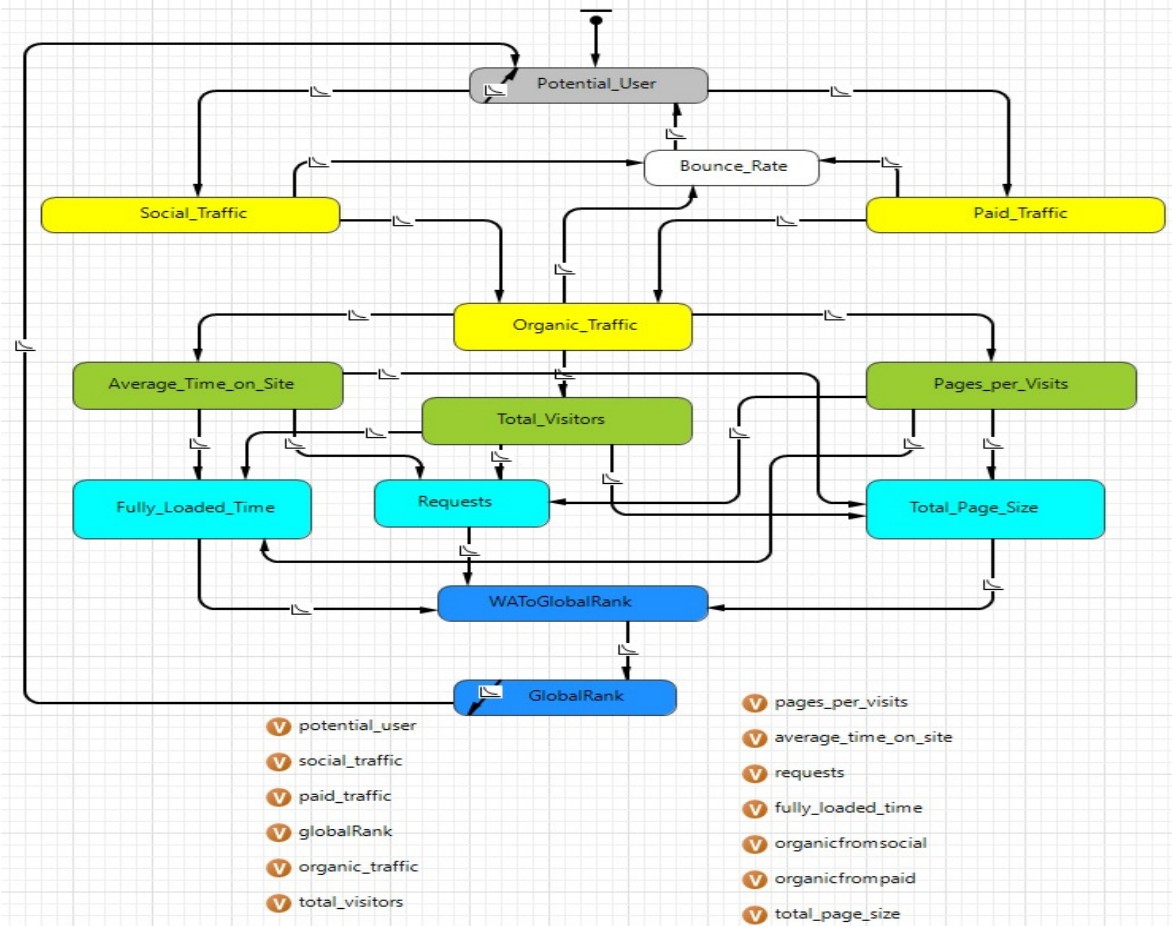

**Figure 5.** Agent-Based simulation model.

The simulation starts from the top gray box where agents are distributed, with the assistance of the statistical analysis, to the logistics website either through paid, social, or organic traffic (yellow boxes). Following, a part of the agents exits the website through the bounce rate white box, and another part continues to the website providing the well-known behavioral metrics, average time on site, total visitors, and pages per visit (green boxes). After that, the agents moved to the technical factors following the Poisson distribution. The technical factors are presented in Figure 5 in cyan boxes. Finally, the agents arrive at the global ranking box (blue), which provides the global rank, and then proceed to the general agents' pool on the top (gray box).

Figure 6a,b illustrate the population allocation results over a period of 180 days. Figure 6a illustrates the 12th day of the simulation, where the gray agents represent the potential visitors and the yellow agents the traffic sources, either social, paid, or organic traffic, of the logistics websites. Figure 6b presents the simulation after the 71st day and

illustrates that more cyan agents are created that represent the technical factors and more blue agents that represent the global rank. This is expected behavior because, after 70 days, the brand name has been created. Additionally, the red agents represent the bounce rate, and the green agents represent the behavioral factors. The following Figures illustrate the results of the simulation.

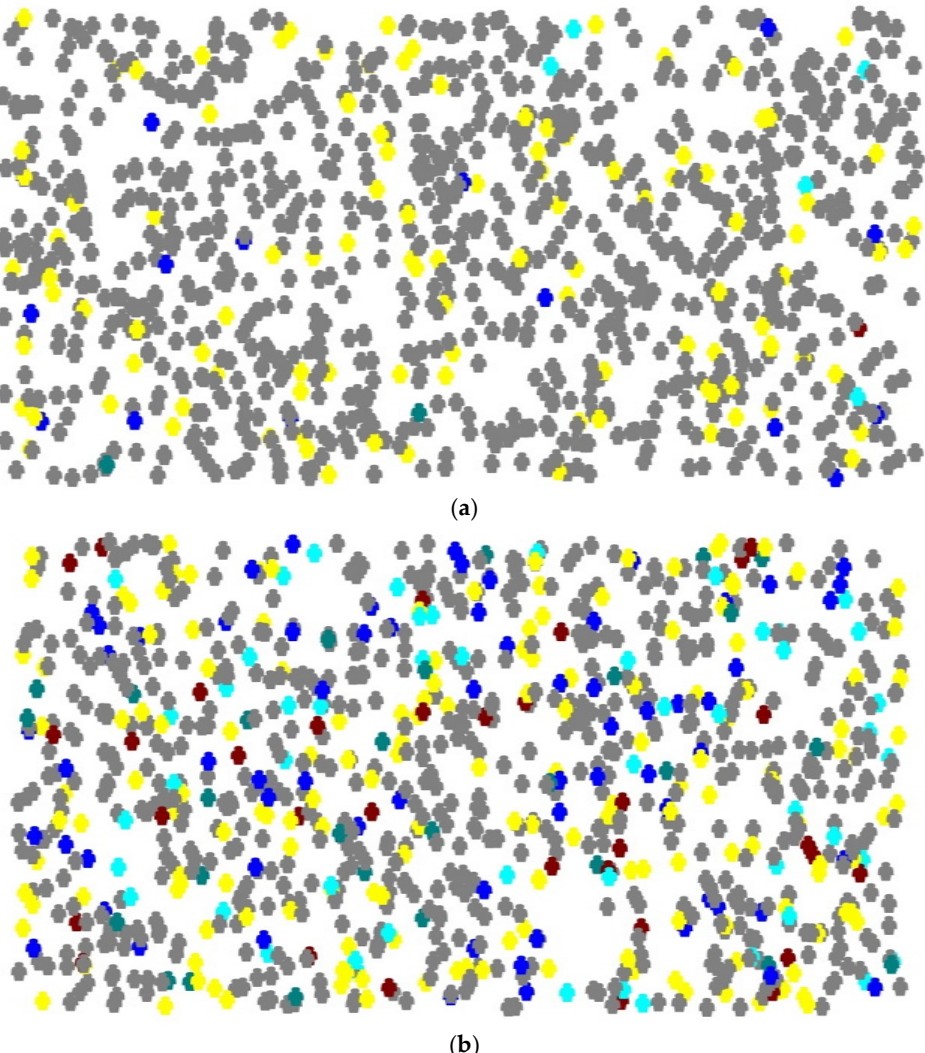

(a)

(b)

**Figure 6.** (**a**) Simulation's population allocation with 10,000 agents during 180 days. Day 12. (**b**) Population allocation of 10,000 agents during 180 days. Day 71.

Figures 7 and 8 depict the simulation's results of the visibility factors, and more specifically, Figure 8 illustrates the total visitors and the organic traffic, and Figure 7 presents the social traffic and paid traffic. The horizontal axis represents the duration of the model for 180 consequent days, and the vertical axis illustrates the values given from the simulation. The yellow line in Figure 7 shows a stream of advertisements (paid traffic) placed from the logistics companies, and the pink line depicts the visitors that access the websites from social media advertisements. In Figure 8, it can be observed that the paid advertisements in Figure 7 have slightly affected the number of visitors to the logistics websites. On the other hand, the placements of social media advertisements in Figure 7 had a crucial impact on the total number of visitors. This interesting result illustrates that the logistics sector websites are affected much more by social media advertisements than the general Google or website advertisements and suggests to logistics industry marketers to invest much more in social media than in other sources to promote visibility.

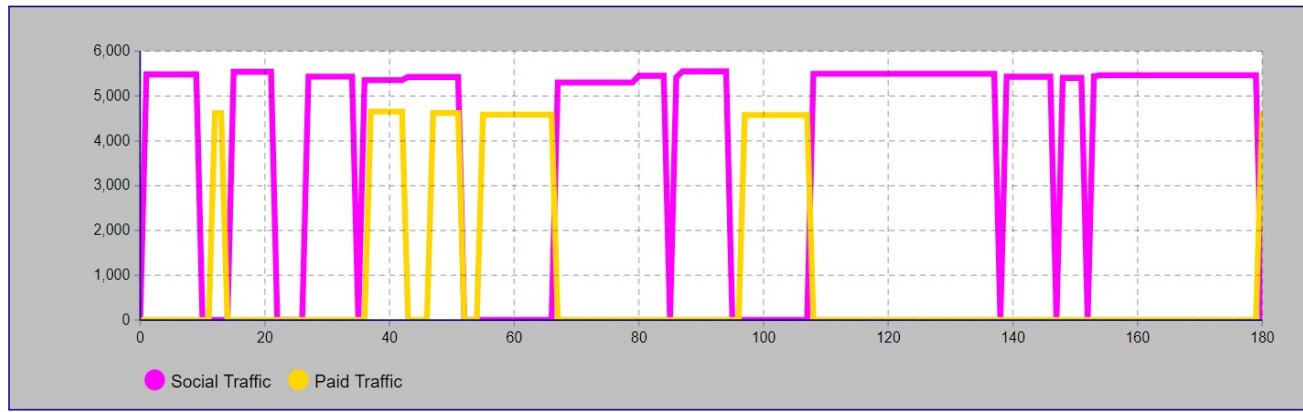

**Figure 7.** The time chart depicts the history of the contribution of the following traffic factors: Social traffic and Paid Traffic during 180 days.

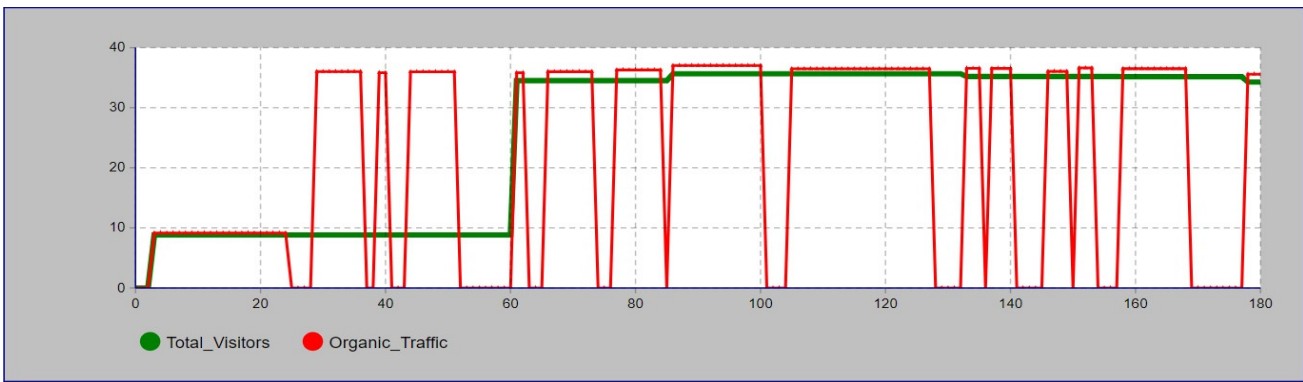

**Figure 8.** The time chart depicts the history of the contribution of the following visibility factors: Organic traffic and Total visitors during 180 days.

Figure 9 depicts the simulation results for the global rank variable. As can be identified, there is a significant improvement when social media advertisement is placed. The major change can be observed again after day 23 and the spike of the social media advertisements, where the global ranking gets much better values day after day (negative values because the 2nd place in the global ranking is better than the 150th place). This finding highlights to decision-makers the importance of social media to the logistics industry as well as the possible beneficial impact of investing in social media in order to optimize the corporate brand name [89]. Finally, the global ranking seems to be impacted much more by social media advertisements than the general paid traffic which means that the social media advertisements have a better return on investment than the general paid advertisements (Google ads or ads on other websites).

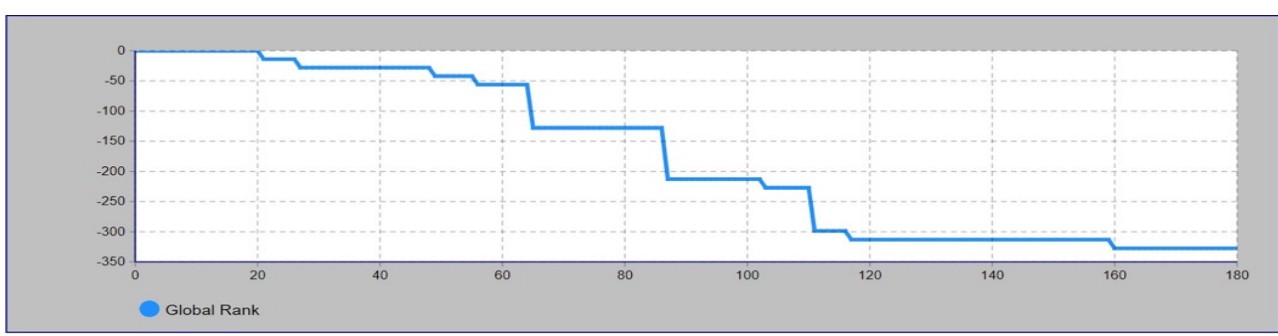

**Figure 9.** The time chart depicts the history of the contribution of the global rank during 180 days.

## 4. Discussion

The main goals of this study are, on the one hand, to examine the effects of the websites' technical factors and behavioral factors on the logistics websites' visibility and brand name, and on the other hand, to suggest digital marketing optimization strategies in order to improve corporate brand name. More specifically, the first hypothesis (H1), which is built on previous studies [90,91], highlights the importance of developing websites with low size to optimize visibility, as presented in the technical optimization scenario in Section 3.2.1. The second hypothesis (H2) illustrates the necessity of placing more advertisements on social media than on other websites since the return on investment could be higher. This significant result is illustrated in the agent-based model and supported by previous researchers [92,93]. Additionally, the findings, as also presented in the technical optimization scenario in Section 3.2.1, highlight the necessity of minimizing the fully loaded time because if a customer must wait too long for a page to load, they are more likely to exit the website immediately [94,95].

The results of the third hypothesis (H3) revealed no correlation between the request and the percentage of visitors that exit the website but illustrated a great correlation between bounce rate and paid traffic. This finding illustrates that a large number of visitors that entered the website from a paid advertisement exit without viewing anything. This practically means that visitors clicked the advertisement and entered accidentally, and this underlines the finding of the first hypothesis. The fourth hypothesis (H4) found a correlation between the time spent on the website, the visibility, and the paid traffic, as presented in the visibility optimization scenario in Section 3.2.1, and on the execution of the agent-based model in Section 3.3, the paid advertisements (social and general) have a beneficial impact on corporate websites' traffic and visibility [14,16,92,93]. Finally, the findings for the fifth hypothesis (H5) were very interesting. Statistical analysis and the brand name optimization scenario illustrated a great impact of technical factors on the corporate global ranking. More specifically, when the website takes less time to load, it increases the ranking and the corporate brand name. Companies must construct user-friendly websites to increase their brand name. For example, the consumer must be able to find a tracking number as fast as possible and without waiting for a webpage to load, which will contribute to the development of brand loyalty [96]. An additional finding highlights the beneficial effects of social media advertisements on the brand name [97,98].

## 5. Conclusions

### 5.1. Theoretical Implications

This research contributed by developing a three-stage data-driven technique for measuring the effect of technical and behavioral factors on seven logistics companies' websites. Additionally, the study broadens researchers' toolbox while attempting to extend crucial digital marketing previous results and future strategies. More specifically, this study presented the usefulness of Fuzzy Cognitive Maps in the examination of macro-level situations and the creation of optimum digital marketing scenarios. Additionally, the adoption of agent-based models provides the researchers with the ability to run iconic simulations for free. This research perspective moves digital marketing research forward to a new and more practical approach.

### 5.2. Practical Implications

The practical implications of this study are three-fold. First, marketers can benefit from the implementation of both general paid and social media advertisements, but they have to emphasize social media advertisements because it will increase the website's visibility and brand name faster. Second, developers need to make sure that the logistics website will be easy to use and user-friendly since the technical factors are highly related to the corporate brand name. Third, decision-makers need to consider the adoption of Fuzzy Cognitive Maps and Agent-based models while designing their corporate strategy. The adoption of

those methods will save cost, effort, time, and potential damage for the companies, since there is no need to test the scenarios in real life but on a platform instead.

### 5.3. Future Research and Limitations

This study implemented an innovative methodology and provided useful insights for logistics managers globally. There are several limitations and suggestions for future research. The current research is based on the seven biggest world-leading logistics websites and provides global practical implications. The next step must be based on microenvironments. For instance, the competition among the logistics websites in the Mediterranean Sea [99]. This will give more accurate findings for SMEs. Another limitation is that there is no in-depth examination of the above advertisements either on social media or on websites. Future research can examine the optimal website advertisements for social media based on neuromarketing [100]. Finally, research data are extracted from data platforms which, on the one hand, provide accurate and indisputable big data, but on the other hand, the organizational culture cannot be identified. Finally, future research could examine and correlate both big data and qualitative research through interviews to reveal underlying mechanisms [101].

**Author Contributions:** Conceptualization, D.P.R., D.P.S., P.T., M.C.T. and C.V.; methodology, D.P.R., D.P.S. and P.T.; software, D.P.R., D.P.S. and P.T.; validation, D.P.R., D.P.S. and P.T.; formal analysis, D.P.R.; investigation D.P.R., D.P.S. and P.T.; resources, D.P.R., D.P.S. and P.T.; data curation, D.P.R., D.P.S. and P.T.; writing—original draft preparation, D.P.R. and M.C.T.; writing—review and editing, D.P.R.; visualization, D.P.R., D.P.S. and P.T.; supervision, D.P.S. and P.T.; project administration, D.P.R., D.P.S. and P.T. All authors have read and agreed to the published version of the manuscript.

**Funding:** This research received no external funding.

**Institutional Review Board Statement:** Not applicable.

**Informed Consent Statement:** Not applicable.

**Data Availability Statement:** Not applicable.

**Conflicts of Interest:** The authors declare no conflict of interest.

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
