# Peer review of "The Effects of Logistics Websites’ Technical Factors on the Optimization of Digital Marketing Strategies and Corporate Brand Name"

_processes, doi:10.3390/pr10050892_

Round 1
Reviewer 1 Report
Content
- The authors constantly refer to the seven world-leading logistics companies that they considered in their research. Although it is understandable why they may not want to disclose the names of these companies, it would be desirable to tell us a little more about them in the Introduction (before stating the key performance indicators). For example, it would be useful to know if these companies are among the top 100 logistics companies, or among the top 10. Are they retail logistics companies? Are they third-party logistics companies? How were they chosen? Why did they choose seven (instead of 5 or 10)? Even if the authors can generalize their results to any other companies, it is important to understand the sources of the data as much as possible.
- In the Introduction (Subsection 1.2 -- near Line 106), the authors talk about audio and video analytics. While these are very interesting topics, they are not mentioned again in the manuscript. It is unclear why they are mentioned in the Introduction if they are not discussed later on. We do not need to know about Vimeo and Dailymotion (unless the authors actually used these video platforms). I suggest removing all the text referring to audio and video analytics, so that the manuscript can concentrate exclusively on what is relevant. If the authors actually performed audio and video analytics as part of their research, then this needs to be explained further.
- Social traffic is widely discussed in the manuscript. However, the definition is vague. The authors state that social traffic corresponds to users redirected from "Facebook, Instagram or social media in general". Did the authors use any social media platform apart from Facebook and Instagram? If so, they need to be clear and specify which other platforms were considered. I realize that the same approach that is applicable to Facebook (for example) can be applied to other platforms. However, did the authors use data from other platforms? If so, which ones? If the authors have not investigated other platforms yet, they must indicate that their work is limited to Facebook and Instagram, though they expect the same approach to be useful and applicable to other social media platforms.
- I recommend that SEO should be removed as a keyword from the list of keywords for this manuscript (Line 33). I appreciate that SEO is a relevant concept, but the acronym appears only once in the text of the manuscript (Line 351). Hence, I do not think it is accurate to claim SEO as a keyword. This submission reports on work about logistics, web analytics, and other topics indicated in the list of keywords. There is no point in enlarging the list with acronyms that are not sufficiently discussed in the text.
Figures
- Figure 4 (Line 408) has a typo. It states "Socia_Traffic", rather than "Social_Traffic". This must be fixed.
Presentation and style
The quality of the presentation of this submission, in terms of English language and style, needs to be improved. There are several typos. In addition, the writing style can be improved considerably. For example, the phrase "in order to" is used more than twenty times in the manuscript (sometimes in the same paragraph). When used excessively, this phrase becomes inappropriate. I suggest avoiding it as much as possible (because it has been overly used).
I can make several suggestions to improve the text (please, read below some of them). However, I recommend that the text should be proofread properly before resubmitting it again.
Line 20: The text should be changed to "depend on their websites to acquire new customers...".
Line 24: Remove "very". The text should be changed to "The first phase of the...".
Line 33: Do the authors mean "bid data" as the first keyword? I think they mean "big data". Please, check and correct if necessary.
Line 46-47: I suggest changing the text to "... the authors attempt to answer these questions based...". Hence, replace "contribute to" with "answer".
Line 50: Replace "in order to" with "to".
Line 53-54: Add "the" before "internet". The text should read: "... the gap among markets and businesses is the internet...".
Line 60-61: Replace "For reaching the..." with "To reach the...".
Line 66: Change the text to "... as well as leads to increase sales and brand name".
Line 69: The paragraph should start with: "This paper is divided into...". Remove "Following that,".
Line 76: Add "is" before "growing". The text should be: "Big data analytics is growing more...".
Line 84: Replace "in order to" with "to".
Line 90: Replace "in order to" with "to".
Line 94: Add "one" after "important". The text should change to "a corporate point of view is the most important one".
Line 98: Remove "been". The text should be "the following questions have emerged".
Line 99: Replace "doesn't" with "does not".
Line 105: Replace "in order to" with "to".
Line 114: Replace "in order to" with "to".
Line 418: Replace "figure" (lowercase) with "Figure" (initial capital).
Line 446 to 456: The word "Figure" must be used uniformly across the entire manuscript. It should always be spelled with the initial capitalized, and it should not be abbreviated (do not use "Fig"). So, in Line 447 replace "fig. 7" with "Figure 7", "fig. 6" with "Figure 6" and so on.
Line 512: I think the authors meant to say that other researchers can use their work to perform simulations for free. Hence, the phrase "without using funds" should be replaced with "for free". Please, confirm if this is what you mean before making any changes.
Line 529: I suggest replacing "The next research" with "The next step".
Line 535: Do the authors mean "bid data"? Or do they mean "big data"? Please, check and correct accordingly.
Author Response
Reviewer 1
Dear Reviewer,
Thank you very much for your detailed review and for the time you spent on our manuscript.
We have addressed all your suggestions in the text.
Kind Regards,
Dimitrios P. Reklitis
Responses
|
The authors constantly refer to the seven world-leading logistics companies that they considered in their research. Although it is understandable why they may not want to disclose the names of these companies, it would be desirable to tell us a little more about them in the Introduction (before stating the key performance indicators). For example, it would be useful to know if these companies are among the top 100 logistics companies, or among the top 10. Are they retail logistics companies? Are they third-party logistics companies? How were they chosen? Why did they choose seven (instead of 5 or 10)? Even if the authors can generalize their results to any other companies, it is important to understand the sources of the data as much as possible. |
We used 7 logistics websites from the 10 top logistics websites based on Statista [49] in order to keep anonymity. Additionally, the research is based on big data and the 7 logistics websites provide sufficient data to perform the research. Addressed in text: lines 144-150 |
|
In the Introduction (Subsection 1.2 -- near Line 106), the authors talk about audio and video analytics. While these are very interesting topics, they are not mentioned again in the manuscript. It is unclear why they are mentioned in the Introduction if they are not discussed later on. We do not need to know about Vimeo and Dailymotion (unless the authors actually used these video platforms). I suggest removing all the text referring to audio and video analytics, so that the manuscript can concentrate exclusively on what is relevant. If the authors actually performed audio and video analytics as part of their research, then this needs to be explained further. |
We deleted all the definitions of audio, text and video analytics and their references. Lines:128-129 |
|
Social traffic is widely discussed in the manuscript. However, the definition is vague. The authors state that social traffic corresponds to users redirected from "Facebook, Instagram or social media in general". Did the authors use any social media platform apart from Facebook and Instagram? If so, they need to be clear and specify which other platforms were considered. I realize that the same approach that is applicable to Facebook (for example) can be applied to other platforms. However, did the authors use data from other platforms? If so, which ones? If the authors have not investigated other platforms yet, they must indicate that their work is limited to Facebook and Instagram, though they expect the same approach to be useful and applicable to other social media platforms. |
The research is limited to Facebook and Instagram. Addressed in text: Table 1 “When a user is redirected from Facebook, Instagram or social media in general to the corporate website, produces the KPI Social Traffic. [60, 64]. This research is limited to Instagram and Facebook.” |
|
I recommend that SEO should be removed as a keyword from the list of keywords for this manuscript (Line 33). I appreciate that SEO is a relevant concept, but the acronym appears only once in the text of the manuscript (Line 351). Hence, I do not think it is accurate to claim SEO as a keyword. This submission reports on work about logistics, web analytics, and other topics indicated in the list of keywords. There is no point in enlarging the list with acronyms that are not sufficiently discussed in the text. |
SEO Removed Lines: 33-34 |
|
Figure 4 (Line 408) has a typo. It states "Socia_Traffic", rather than "Social_Traffic". This must be fixed. |
Changed the figure Addressed in text: Figure 4 |
|
The quality of the presentation of this submission, in terms of English language and style, needs to be improved. There are several typos. In addition, the writing style can be improved considerably. For example, the phrase "in order to" is used more than twenty times in the manuscript (sometimes in the same paragraph). When used excessively, this phrase becomes inappropriate. I suggest avoiding it as much as possible (because it has been overly used). |
We improved the language and removed the majority of "in order to" in the text |
|
I can make several suggestions to improve the text (please, read below some of them). However, I recommend that the text should be proofread properly before resubmitting it again. Line 20: The text should be changed to "depend on their websites to acquire new customers...". (DONE) Line 24: Remove "very". The text should be changed to "The first phase of the...". (DONE) Line 33: Do the authors mean "bid data" as the first keyword? I think they mean "big data". Please, check and correct if necessary. (DONE) Line 46-47: I suggest changing the text to "... the authors attempt to answer these questions based...". Hence, replace "contribute to" with "answer". (DONE) Line 50: Replace "in order to" with "to". (DONE) Line 53-54: Add "the" before "internet". The text should read: "... the gap among markets and businesses is the internet...". (DONE) Line 60-61: Replace "For reaching the..." with "To reach the...". (DONE) Line 66: Change the text to "... as well as leads to increase sales and brand name". (DONE) Line 69: The paragraph should start with: "This paper is divided into...". Remove "Following that,". (DONE) Line 76: Add "is" before "growing". The text should be: "Big data analytics is growing more...". (DONE) Line 84: Replace "in order to" with "to".(DONE) Line 90: Replace "in order to" with "to". (DONE) Line 94: Add "one" after "important". The text should change to "a corporate point of view is the most important one". (DONE) Line 98: Remove "been". The text should be "the following questions have emerged". (DONE) Line 99: Replace "doesn't" with "does not". (DONE) Line 105: Replace "in order to" with "to". (DONE) Line 114: Replace "in order to" with "to". (DONE) Line 418: Replace "figure" (lowercase) with "Figure" (initial capital). (DONE) Line 446 to 456: The word "Figure" must be used uniformly across the entire manuscript. It should always be spelled with the initial capitalized, and it should not be abbreviated (do not use "Fig"). So, in Line 447 replace "fig. 7" with "Figure 7", "fig. 6" with "Figure 6" and so on. (DONE) Line 512: I think the authors meant to say that other researchers can use their work to perform simulations for free. Hence, the phrase "without using funds" should be replaced with "for free". Please, confirm if this is what you mean before making any changes. (DONE) Line 529: I suggest replacing "The next research" with "The next step". (DONE) Line 535: Do the authors mean "bid data"? Or do they mean "big data"? Please, check and correct accordingly.(DONE)
|
The text has been proofread as advised and every single change has been addressed in the text. |
Reviewer 2 Report
- Please enhance the research gaps and unique perceptiveness of this study.
- Study purposes could be more identified and described clearly.
- In Table 6, 10 12, the standardized coefficients those p <0.001 should marked two start. (Maybe missed)
- In Table 12, although F value is significant, the R2 only 0.107, the conclusion of H5 be interpreted reluctantly.
Author Response
Reviewer 2
Dear Reviewer,
Thank you very much for your detailed review and for the time you spent on our manuscript.
We have addressed all your suggestions in the text.
Kind Regards,
Dimitrios P. Reklitis
Responses
|
Please enhance the research gaps and unique perceptiveness of this study |
Addressed in text: lines 213-220 |
|
Study purposes could be more identified and described clearly |
Addressed in text: lines 264-270 |
|
In Table 6, 10 12, the standardized coefficients those p <0.001 should marked two start. (Maybe missed) |
Addressed in text: Table 6 à line 361 Table 10 à line 401 Table 12à line 422
|
|
In Table 12, although F value is significant, the R2 only 0.107, the conclusion of H5 be interpreted reluctantly. |
Addressed in text: lines 431-432 |